# Novel Therapeutic Strategies in the Treatment of Systemic Sclerosis

**DOI:** 10.3390/ph16081066

**Published:** 2023-07-27

**Authors:** Olga Gumkowska-Sroka, Kacper Kotyla, Ewa Mojs, Klaudia Palka, Przemysław Kotyla

**Affiliations:** 1Department of Rheumatology and Clinical Immunology, Voivodeship Hospital No. 5 in Sosnowiec, Medical University of Silesia, 40-055 Katowice, Poland; oag@poczta.onet.pl; 2Department of Internal Medicine Rheumatology and Clinical Immunology, Medical University of Silesia, 40-055 Katowice, Poland; kacper.kotyla@gmail.com (K.K.); klaudiamuszalik01@gmail.com (K.P.); 3Department of Clinical Psychology, Poznan University of Medical Sciences, 61-701 Poznan, Poland; ewa_mojs@poczta.onet.pl

**Keywords:** systemic sclerosis, treatment, Janus kinase inhibitors, fibrosis, anti-cytokine treatment

## Abstract

Systemic sclerosis is a connective tissue disease of unknown origin and with an unpredictable course, with both cutaneous and internal organ manifestations. Despite the enormous progress in rheumatology and clinical immunology, the background of this disease is largely unknown, and no specific therapy exists. The therapeutic approach aims to treat and preserve the function of internal organs, and this approach is commonly referred to as organ-based treatment. However, in modern times, data from other branches of medicine may offer insight into how to treat disease-related complications, making it possible to find new drugs to treat this disease. In this review, we present therapeutic options aiming to stop the progression of fibrotic processes, restore the aberrant immune response, stop improper signalling from proinflammatory cytokines, and halt the production of disease-related autoantibodies.

## 1. Introduction

Systemic sclerosis (SSc) is a connective tissue disease of an autoimmune background characterized by the dysregulation of the immune response, and manifested as altered T- and B-lymphocyte function, autoantibody synthesis, and vascular bed damage, leading to peripheral tissue hypoxia and massive tissue fibrosis [1]. As a result, patients with systemic sclerosis experience progressive endothelial injury with abundant collagen deposition in the skin and internal organs, resulting in terminal organ dysfunction. With a prevalence of 7.2–33.9 and 13.5–44.3 per 100,000 individuals in Europe and North America, respectively, and an annual incidence of 0.6–5.6 per 100,000 individuals, SSc is recognized as a rare disease [2]. The disease is characterized by high clinical heterogeneity varying from no skin lesions (scleroderma sine scleroderma) to massive skin involvement (diffuse systemic sclerosis). Several subtypes of the disease have been described. The most common classification is one proposed by Le Roy who distinguished diffuse and limited subsets of systemic sclerosis on the basis of skin involvement. In limited systemic sclerosis, fibrosis is restricted to the hands, face, and feet. In contrast, in the generalized form, skin sclerosis extending proximal to the elbows and knees can be seen and may involve truncal areas [3].

The cause of the disease is still a topic of debate among scientists, with many potential mechanisms being suggested. Patients with systemic sclerosis (SSc) almost universally have antinuclear antibodies in their blood, along with specific autoantibodies based on the disease subset [4]. This autoantibody presence can be observed in 95% of cases at the initial diagnosis [5], which is used to identify the preclinical stage of the disease known as VEDOSS (very early diagnosis of SSc). ANA positivity is one of three red flags, along with puffy fingers and Raynaud’s phenomenon [6]. The disease has two major subtypes based on the extent of skin involvement, as well as other types, such as CREST syndrome, overlap syndrome, and systemic sclerosis without skin involvement.

This review attempts to systematically assess the utility of dozens of various compounds for SSc treatment started in the 1960s. Since that time, it has been apparent that no specific drug that is able to halt the disease progression currently exists [7]. Therefore, special emphasis has been put on treating scleroderma-related internal organ involvement, and this philosophy currently guides treatment. Organ-based treatment aims to stop the progression of internal organ damage and preserve their function [8].

## 2. EULAR Recommendations for Treatment Patients with SSc

EULAR recommendations for the treatment of systemic sclerosis published in 2017 [9] address the therapeutic approaches in several SSc-related organ complications such as the following: Vascular disease (Raynaud’s phenomenon and digital ulcers);Pulmonary arterial hypertension (PAH);Skin fibrosis;Interstitial lung disease;Scleroderma renal crisis;Gastrointestinal involvement.

However, an unmet need that remains is to identify a systemic treatment that may slow or stop the progression of the disease. Decades after being promised, progress was on the way, and SSc research started taking tangible steps towards testing findings from laboratory studies in clinical settings. Based on the advances in rheumatology and clinical immunology, a therapeutic approach focused on reducing disease activity as a whole may be an option rather than only preventing damage to one particular type of cells. This includes the following:Reducing/halting the activity of fibrotic processes;The restoration of the proper functioning of immunocompetent cells;Targeting specific cytokines;A reduction in antibody synthesis.

## 3. Skin and Visceral Organ Fibrosis

Fibrosis of the skin and internal organs is a hallmark of systemic sclerosis. The precise mechanism leading to uncontrolled fibrosis is only partially understood, and many theories exist. However, none of them satisfactorily explain all phenomena occurring during the course of the disease. At the current level of understanding, the fibrotic process is the direct result of aberrant immunocompetent cell function. However, this process is directly mediated by several “fibrotic messengers“ and cytokines activating a plethora of profibrotic pathways [10,11].

There are several fibrogenic factors; among them are platelet-derived growth factor (PDGF), tumour necrosis factor-α (TNF-α), and transforming growth factor-beta (TGF-β), which are believed to be the most important.

### Transforming Growth Factor-β

TGF-β is a pleiotropic cytokine involved in promoting wound repair and suppressing inflammation. Having potent profibrotic properties, TGF-β activity has been linked with several fibrosis-related disorders such as pulmonary fibrosis, glomerulosclerosis, renal interstitial fibrosis, cirrhosis, Crohn’s disease, cardiomyopathy, scleroderma, and chronic graft-versus-host disease [12,13]. TGF-β exists in three isoforms: TGF-β1, TGF-β2, and TGF-β3. They share a similar biologically active region and use the same type I and type II TGF-β receptor complex for signalling. The most abundant isoform seen in humans is TGF-β1 that is expressed by most human cells. However, it seems that the main source of TGF-β is platelets, which have a moderate concentration of TGF-β but are present in the blood in high numbers [14]. TGF-β exerts a potent stimulatory effect on fibroblasts and matrix synthesis; thus, it is commonly recognized as a pivotal cytokine responsible for driving the fibrotic process [15]. The TGF-β receptor is formed by type II receptor homodimers (TβRII, ACTRII, AMHRII or BMPRII) coupled with a type I receptor (activin-like kinase 1–7) and stabilized by a co-receptor (also known as type III receptor—betaglycan and endoglin). After binding the ligand to the TGF, receptor phosphorylation of the SMAD molecules occurs, leading directly to the activation of the SMAD4 molecule which translocates to the nucleus and regulates the expression of targeted genes. Apart from this canonical way of signalling, TGF-β can activate several mitogen-activated protein pathways such as c-Jun N terminal kinase (JNK) and extracellular signal-regulated kinase (ERK). This causes the subsequent activation of Ras, Raf, ERK and mitogen-activated protein kinase 1/2, which eventually activate SMAD and transcription factors in the nucleus. This leads to many pathophysiological consequences. As established recently, there is a close association between GF-β /Ras-ERK1/2 signalling and activation of the second main player in SSc pathogenesis—namely endothelin-1 (ET-1). The role of TGF-β has been intensively studied in various SSc models. With the current level of knowledge, it is speculated that TGF acts locally rather than systemically. A recent study in patients with SSc showed that all three isoforms of TGF-β were present in lower concentrations in SSc patients than in healthy subjects, but the cytokine was shown to be hyperexpressed in dermal fibroblasts [13]. On the other hand, TGF-β is a potent stimulator of the epithelial–mesenchymal transition in vitro [16]. Several key questions remain unanswered. How will these findings translate to the development of novel therapeutic strategies? Is the TGF blockade the proper way to stop fibrosis? How can it be realized? 

A reduction in TBF-β signalling can be realized by the modulation of at least four ways of TGF beta activity:Blocking TGF synthesis;Blocking the ligand;Blocking latent activation;Blocking intracellular signalling.

Several studies have addressed these pathways in in vitro and preclinical studies with promising results. A topical application of a peptide inhibitor of TGFβ_1_ (P144) in a mouse model of systemic sclerosis revealed the remarkable suppression of connective tissue growth factor expression in fibroblast SMAD2/3 phosphorylation, and α-smooth muscle actin positive myofibroblast development [17]. Recently, in a similar animal model, LG283, a curcumin derivate, suppressed fibrosis in mice as well as in cultured human fibroblasts, antagonizing the TGF-β/SMAD3 pathway [18]. Parallel to this, a Chinese group addressed the potential of Iguratimod (T-614), a novel DMARD currently being tested for use in rheumatoid arthritis in China and Japan [19,20]. According to the results obtained from the study, Iguratimod negatively regulated the TGF-β1/Smad pathway and inhibited TGF-β1-induced dermal fibroblast proliferation [21]. Another therapeutic approach has recently been proposed by Jiang et al. They tested polyporus polysaccharide (PPS) Chinese herb derivates on the TGF-β1/ Smad2/3 pathway. They showed that PPS exerts potent antifibrotic effects by inhibiting fibroblast-to-myofibroblast transition, halting extracellular matrix deposition, and modulating lung fibroblast proliferation and migration [22]. Unfortunately, these promising results have not been fully confirmed in human studies.

The first study testing recombinant human anti-transforming growth factor β_1_ antibody in patients with systemic sclerosis, completed in 2007, did not show evidence of efficacy [23]. More promising data came from the study of Rice et al., which showed that Fresolimumab, a high-affinity neutralizing antibody targeting all three TGF-β isoforms, significantly improved skin disease with a concomitant reduction in dermal myofibroblast infiltration, and lowered the expression of the gene thrombospondin-1 (*THBS1*), a biomarker of SSc skin disease highly upregulated by TGF-β (Figure 1).

TGF-β is believed to play a key role in driving the fibrotic process in systemic sclerosis. The amelioration of its activity may translate to the halting of the fibrotic process in the disease. This can be realized by the use of P144, which is able to block the interaction of TGF-β with its membrane receptors or by blocking the TGF molecule (Fresolimumab). With the use of Iguratimod, the blockade of SMAD activation can be achieved. The direct activation of targeted genes in the nucleus can be stopped by the interruption of SMAD signalling with the use of novel molecules, such as LG283 or PP5

## 4. The Other Anti-Fibrotic Treatments

### 4.1. Oncostatin M

Oncostatin M (OSM) is a member of the interleukin-6 family that signals through the JAK/STAT system. OSM levels were found to be elevated in the serum of SSc patients. Moreover, the skin of SSc patients is characterized by a high expression of the OSM receptor, suggesting the pathophysiological role of this cytokine in the development of the disease [24,25]. Unfortunately, a recently completed trial with GSK233081—a humanized IgG1 antibody targeting the OSM receptor—failed to show any therapeutic effect in SSc [26].

### 4.2. Rho-Associated Kinases

The other potential anti-fibrotic therapeutic approach is to modulate Rho-associated kinase activity. Rho kinases (ROCK) are essential downstream effectors of the Rho GTPases and are responsible for important physiological functions, such as the organization of actin, cytoskeletal cell cycle control, apoptosis, and the regulation of cell–cell adhesion [27]. It has been established that ROCKs are crucial for myofibroblast differentiation and the synthesis of the extracellular matrix. Thus, this could be a potential therapeutical target for SSc-related fibrosis. In line with this, Belumosudil, a ROCK inhibitor licensed for the treatment of chronic graft-versus-host disease, is under investigation for systemic sclerosis in the USA (NCT02841995 and NCT04930562) [28].

## 5. Vasculopathy

Vasculopathy is a term that was almost exclusively used for the description of small and large vessel abnormalities in SSc [29]. This process is commonly recognized as the third (among fibrosis and immune system dysfunction) key element in the pathogenesis of systemic sclerosis. Vascular injury is believed to be initiated by the influence of environmental factors and autoimmune responses directed toward endothelial cells [30]. The latter is caused mainly by anti-endothelial antibodies. Vascular abnormalities seen in SSc are classified into two categories: (1) destructive vasculopathy and (2) impaired vasculogenesis and angiogenesis. In the early stages of SSc, the predominant activity of inflammatory cells leads directly to the activation of endothelial cells (ECs) [31]. Parallel to this, the increased synthesis of pro-angiogenic factors (mainly VEGF and endothelin-1) is observed despite the impaired response of ECs to those factors. At the functional level, vasculopathy is characterized by excessive reactive oxygen species formation, the infiltration of Th cells polarized toward a Th2 and Th17 response, the local infiltration of macrophages, and the activation of an endothelial-to-mesenchymal transition promoting vascular remodelling and tissue fibrosis [32].

## 6. Immunocompetent Cells in Systemic Sclerosis

### 6.1. B Cells in Systemic Sclerosis

The role of B cells in the pathogenesis of autoimmune diseases has been conclusively confirmed. In systemic sclerosis, the dysregulation of the immune system and breaking the tolerance towards nuclear antigens are currently recognized as hallmarks of the disease [33]. The role of B cells is not, however, restricted to the production of scleroderma-related antibodies, which seem to play an indicative rather than causative role in SSc [34]. B cells are a rich source of potent proinflammatory cytokines, which may also act as antigen presenting cells, co-stimulate T cells and synthesize profibrotic agents [35]. Thus, targeting the B cell population seems to be a reasonable therapeutic approach [36].

With many therapeutic approaches targeting B cells with anti-CD20 antibodies, therapies which involve the targeting of specific B cell populations (anti-CD38, anti-CD22), modulation of intracellular B cell signalling pathways, co-stimulation blockade (anti-CD40), and the targeting of B cell survival factors (anti-BAFF), or more recently introduced cellular immunotherapy approaches (anti-CD19 CAR T cells), have all attracted special attention [36]. Recently, a case report on CD19-CAR T cells showed promising data on the utility of such a treatment in patients with severe systemic sclerosis [37].

### 6.2. Anti-CD20 Therapy

Rituximab (RTX), a chimeric anti-CD20 monoclonal antibody, is one of the best-studied therapeutic compounds in systemic sclerosis. It targets the CD20 molecule expressed on cells of pre-B to pre-plasma cell stages. As a result, treatment with RTX results in long lasting therapeutic effects. The role of RTX in the treatment for SSc has been reported in two small randomized controlled trials, in 16 patients versus placebo, and in 60 patients versus cyclophosphamide in patients with early SSc (<2 years duration). They showed an improvement in the FVC and a reduction in the modified Rodnan Skin Score (mRSS) in the RTX group. However, these results should be interpreted with caution, as the small number of patients included may be linked with statistical bias [38,39]. The results from all small trials, non-controlled and controlled studies were recently summarized by Goswani et al. [40]. In their meta-analysis, they showed significant improvements in lung function in systemic sclerosis-related lung fibrosis. Additionally, RTX was characterized by a good safety profile, with less infection and adverse events observed in patients treated [40]. Recently, the results from phase 2 trials with Rituximab (RECITAL and DESIRES) showed promising results for skin and lung fibrosis improvement [41,42]. However, a key question remains: will results from these studies be confirmed in larger phase III trials?

### 6.3. T Cells

The second main player in the field of systemic sclerosis is the T cell. T cells have been shown to modulate and drive the development of autoimmunity, inflammatory responses and fibrosis. In the course of the disease, the alteration of the frequencies of lymphocytes is observed, suggesting the pivotal role of these cells in the progression of the disease [43]. The role of T cells in a pathophysiological context in systemic sclerosis can be recognized, as T cells interact with B cells to produce specific autoantibodies as well as being a source of potent proinflammatory cytokines [44]. T helper (Th) cells are involved both in the early and late phase of the disease. Taking into consideration the concept of Th cell polarisation toward a Th1 or Th2 response, it is worth noting that Th1- but also Th17-related cytokines, such as TNFα, IFN, IL-1, IL-12, and IL17 act predominantly at early stages of this disease, being responsible for driving the inflammation [45]. With the progression of the disease, inflammation is less active and Th2 cytokines (IL-4, IL-13, IL-5, IL-6, and IL-10) become more prominent [46,47].

The modulation of T cell function may also be realized indirectly by its interaction with cell-to-cell signalling (co-stimulation). Essential molecules involved in this process are the inducible T cell costimulator ICOS and the CD 28 molecule, that bind the ICOS ligand and CD80/86 ligands, respectively. Acazicolcept, a double-ICOS and -CD28 antagonist, has been tested in a mouse model of systemic sclerosis. Treatment with Acazicolcept induced a significant reduction in dermal thickness, collagen content, myofibroblast number, and immunocompetent cell number (B cells, T cells, neutrophils, and macrophages) in a Fra-2 Tg mouse model. Additionally, Acazicolcept treatment reduced lung fibrosis and right ventricular systolic pressure (RVSP). Moreover, treatment with Acazicolcept resulted in a reduction in the frequency of CD4+ and T effector memory cells and an increase in the percentage of CD4+ T naïve cells in the spleen and lung tissue of animals treated [48].

The other therapeutic option is to target CD30. The CD30 molecule, also known as Ki-1 or TNFRSF8, was first identified in 1982 in patients with Hodgkin (HL) lymphoma. The expression of this molecule is, however, not restricted to Hodgkin and Reed Stenberg cells. CD30 is expressed in a small subset of activated T and B lymphocytes, and a variety of lymphoid neoplasms, with the highest expression in classical HL and anaplastic large cell lymphoma. Subsequent studies have demonstrated that CD30 expression is not uniform across all activated lymphocytes, instead being limited to subpopulations of CD4^+^/CD45RO^+^ and CD8^+^ T cells in lymph nodes and the thymic medulla. CD30 expression appears to be higher in CD4+ and CD8+ cells producing Th2-type cytokines [49,50,51]. Soluble CD30 molecules have been found in high concentrations in the sera of patients with systemic sclerosis, which indirectly suggests the pathogenic role of this molecule and, in a wider context, the Th2 response in the development of the disease [52,53,54,55,56,57]. At the moment, two clinical trials are ongoing to test the safety and efficacy of Brentuximab vedotin (BV), a CD30-directed antibody–drug conjugate, which is already approved by the US FDA for the treatment of classic Hodgkin lymphoma (cHL). The first results in patients with SSc will be available in 2023 (NCT03222492 and NTC03198689).

The full activation of T cells is a multi-step process that requires not only the recognition of antigens by T cells but, also, a second co-stimulatory signal, provided by the binding of the CD28 receptor on T cells to CD80 and/or CD86 molecules on the surface of antigen-presenting cells [58]. At the time of T cell activation, the CD4 cell may express a regulatory molecule, namely, CD152 or Cytotoxic T lymphocyte-associated protein-4 (CTLA-4), an inhibitory receptor expressed constitutively on CD4+CD25+ T regulatory lymphocytes (Treg) and transiently on activated CD4+ and CD8+ T lymphocytes. With a very high affinity to CD28 (150 times more than CD80/86), CD152 is able to halt the activation of T cells, preventing an uncontrolled immune response [59]. That was the theoretical background for the synthesis and introduction of Abatacept, a fusion protein composed of a soluble form of the extracellular domain of CTLA-4 linked to the immunoglobulin G1Fc part [60]. It showed its safety and efficacy in rheumatoid arthritis and is still intensively tested in various autoimmune diseases [61,62,63,64,65,66,67,68]. In systemic sclerosis, which is characterised by the pivotal role of the T cell orchestrating the immune response [69], blocking T cell activation seems to be a reasonable approach. Indeed, data from a small retrospective multicentre study with 27 patients showed the potential utility of Abatacept in the treatment of systemic sclerosis, where a reduction in skin involvement, as well as an improvement in tender and swollen join count, were observed [70]. Following these promising data, the ASSET trial (A Study of Subcutaneous Abatacept to Treat Diffuse Cutaneous Systemic Sclerosis), which was a phase 2, double-blind, placebo-controlled trial of weekly subcutaneous Abatacept over 12 months in patients with early diffuse cutaneous systemic sclerosis (≤36 months of disease), was started. The ASSET trial showed only a moderate (statistically non-significant) improvement with Abatacept in the primary endpoint of the mean change from baseline to month 12 in skin score [71]. An open-label extension phase of this study confirmed the good safety profile of Abatacept; however, again the primary endpoint of change in mRSS was not reached [72]. In spite of not reaching a significant improvement in skin score, abatacept remains a potential candidate as a disease-modifying drug in systemic sclerosis, and more trials are needed to confirm the role of this drug in the treatment of the disease.

### 6.4. Targeting the Specific Cytokines

Skin fibrosis, vasculopathy and inflammation are the three crucial elements of this disease. Each process is precisely orchestrated by cytokine and chemokine activity; thus, the rational approach is to block or (rarely) enhance the function of a given cytokine. Among the many cytokines that have already been identified, only a few of them participate in the initiation and progression of SSc.

It is worth keeping in mind the many limitations of the pure categorisation of autoimmune diseases as Th1- or Th2-dependent conditions. SSc may be categorised, at least partially, as a Th2-dependent disease, with a prominent Th2 immune response and the subsequent release of Th2-dependent cytokines, such as IL-4, IL-5, and IL-13, which are able to control the fibrotic process [73]. 

However, in the early stages of scleroderma Th1 cells and Th17 cells are suggested to dominate the immune profile [74], later shifting to Th2 [75] and this may explain why patients with SSc are characterized by the overexpression of IFNα at the early stages of the disease, which than translates to the disease’s development [76,77]. It shows clearly that both arms of the immune response—cellular and humoral are involved in the pathogenesis of SSc, however their role depends on the stage of the disease. This suggests that the direct therapeutic targeting of a cytokine should be performed in the proper phase of the disease.

Several studies have shown that IL-4, a typical representative of the Th2-dependent response, plays an important role as a profibrotic cytokine that stimulates collagen synthesis by fibroblasts. This was proven in laboratory studies where IL-4 stimulated human dermal fibroblasts to synthesize type I and III collagen and fibronectin [78,79]. High levels of IL-4 were reported in patients with systemic sclerosis, where it plays a pathogenetic role, inducing the formation of the extracellular matrix [80,81].

Even more studies refer to the other typical profibrotic cytokine IL-13. IL-13 has several similarities to IL-4 at the amino acid level, and both cytokines display 20–30% homology [82]. Moreover, they show similar biological activity signalling through receptor heterodimers built with combinations of three possible subunits: IL-4Rα, the common gamma chain (γc), and the IL-13Rα1 common receptor chain [83]. IL-4 and IL-13 share a common type I cytokine receptor composed of IL-4α and gamma chain (γc) receptor alpha (IL-4Rα), coupled with the Janus kinase/signal transducer and the activator of the transcription protein 6 (JAK/STAT6) signalling pathway. 

The other receptor that IL-13 may bind to is IL-13Rα2. IL-13 preferentially binds to this type of receptor with a very high affinity. Binding of this cytokine, however, does not exert any physiological response as this type of receptor is commonly considered a “decoy” receptor, as it has a short cytoplasmic tail with no recognizable signalling motifs [84]. Finally, IL-4 and IL-13 may signal through the type II cytokine receptor, composed of IL-4Rα and Il-13Rα2. IL-4 and IL-13 are commonly recognized as leading cytokines driving the fibrotic process [85,86]. Both cytokines signal through a type I or type II cytokine receptor attached to protein kinases, commonly referred as Janus kinase (JAK), with the subsequent activation of the JAK/STAT pathway. These, finally, control the expression of several genes, resulting in cytokine synthesis and the escalation of the fibrotic process [87,88]. This finding may potentially open new therapeutic approaches. Type I and II cytokines attached to Janus kinases may be simply blocked with the use of small synthetic compounds—Janus kinase inhibitors [89].With a large profibrotic potential, both IL-4 and IL-13 are natural targets to inhibit, aiming to ameliorate the fibrotic process [90,91]. Indeed, the targeting of IL-4 and IL-13 has been tested in various fibrotic diseases, including systemic sclerosis [85]. The main concept of targeting IL-4/IL-13 is based on the fact that these cytokines represent a Th2 response. The reasonable approach for SSc treatment is, therefore, to shift the immune response toward Th1. The IL-4 and IL-13 signalling pathways can be stopped at various levels: (1) by ameliorating the soluble cytokine activity, (2) targeting and inhibiting their receptors on cell surfaces, or (3) blocking their intracellular signalling pathways. The first approach is easily achieved by simply blocking IL-4/IL-13 activity with the use of targeted monoclonal antibodies. Quite recently, data on a randomised, double-blind, placebo-controlled, 24-week, phase II, proof-of-concept study of Romilkimab (SAR156597) in early diffuse cutaneous systemic sclerosis became available. In the study, Romilkimab, a humanised, bispecific immunoglobulin-G4 antibody that binds and neutralises IL-4/IL-13, was tested in patients with the diffuse type of SSc. It demonstrated a significant effect on skin changes in early dsSSc [92]. However, the results of the study should be interpreted cautiously given the small sample size, and this result obviously requires confirmation in a future phase III study. 

The second option is to block the receptor function with the use of a monoclonal antibody. For this, Dupilumab, a monoclonal humanized antibody targeting subunit IL-4Rα, recently commercialized for atopic dermatitis and asthma, seems to be a reasonable approach [93,94]. At this moment, however, Dupilumab has not been tested in systemic sclerosis. 

The last approach is to block the signalling pathway with the use of JAK inhibitors JAKi) [89]. Cytokines are small-molecular-weight transmitters essential in cell-to-cell interactions to modulate the innate and acquired immune responses. Based on the similar structure, they are typically divided into several cytokine families. Cytokines signal via typical receptors which can be categorized into several receptor superfamilies. They interact with the extracellular domain of the receptor, which, when activated, can transmit the signal via long chains of transmission molecules to activate specific genes in the nucleus. Among them, cytokines belonging to the class I and class II receptor family utilize receptors which lack an intracellular enzymatic domain and require specific kinases to fully activate the receptor and enable it to transmit the signal into the nucleus [95]. The Janus kinase family enzymes are non-receptor tyrosine kinases that phosphorylate cytokine receptors, enabling them to start signalling through the JAK-STAT signalling pathway. Considering that JAK-STAT signal transduction is initiated by the binding of ligands, such as cytokines to their receptors, proper JAK activity in the JAK-STAT pathway is a key element to orchestrate the immune response, and dysfunctional JAKs are directly responsible for cancers, immune system-related diseases, and autoimmune disorders [96]. These enzymes, referred to as Janus kinases or simply JAK kinases, are an essential element for the transmission of cytokine signals for such important cytokines as IL-2, IL-4, IL-7, IL-9, IL-15, and IL-21, which transmit their signals via gamma chain receptors (γc) [97], since GM-CSF, IL-3, and IL-5 signalling is via the beta type of receptor [98]. Another class of receptor is the heterodimer composed of the gp 130 protein (or its homologue), which is responsible for signalling from IL- 6, IL-11, IL-31, IL-35, and IL-27 [99]. Finally, the class I receptor subfamily, which has a common p40 subunit, interacts with IL-12 and IL-23 [100]. The second group of receptors, called class II receptors, is responsible for transmitting signals from interferons and the IL-10 cytokine family (IL-10, IL-19, IL-20, IL-22, IL-24, and IL-26) [101].

All types of receptor utilize a combination of the four known JAK kinases (JAK1, JAK2, JAK3 and TYK2). This makes it possible to stop or at least modulate cytokine signalling by blocking JAK activity. This was the theoretical background for the development of JAK inhibitors. Contrary to the large, high-molecular-weight biologics targeting one specific cytokine, JAK inhibitors are small synthetic compounds, easy to synthetize and handle. This group of cytokine modulators have shown their safety and efficacy in several rheumatological, dermatological and haematological conditions, and are being intensively tested in many others including connective tissue diseases and other autoimmune disorders [102,103,104,105,106].

The role of JAK kinase inhibition in SSc patients can be re-explained in the context of the activity of the typical Th2-dependent cytokines IL-4 and IL-13, which are responsible for driving the profibrotic process in the body. It is worth noticing, however, that blocking one JAK may also result in the inhibition of other potentially proinflammatory and profibrotic cytokines, and so creating a strong antifibrotic milieu. 

The other players in this field are the IL-12 family of cytokines. Systemic sclerosis is characterized by the activation of IL-12 cytokine family members, which are responsible for driving the profibrotic effect. Therefore, the inhibition of JAK signalling from IL-12 specific receptors may block IL-12, IL-23 and IL-27, thus ameliorating a significant profibrotic effect. The role of the IL-6 cytokine was already mentioned in the light of biologics (Tocilizumab) tested in SSc. IL-6 activity amelioration may alternatively be blocked by JAK inhibitors. Again, with a single compound, we have the potential to block all IL-6 family members simply due to the similarities in the IL-6 receptor family which are coupled with the JAK kinases.

The role of blocking a cytokine class II receptor in systemic sclerosis is less clear. From a theoretical point of view, class II cytokines (IL-10 family cytokines and interferons) are able to transmit both pro- and anti-inflammatory signals. IL-10 is predominantly produced by Bregs that are able to suppress skin fibrosis [107]. It has also been shown that IL-10-producing Bregs are reduced in patients with SSc and correlate with disease activity, suggesting the anti-inflammatory potential of IL-10 in the disease [108,109]. Less is known about the role of the other IL-10 family cytokines, such as IL-20 or IL-23, in the development of systemic sclerosis. Data from the literature suggests the reduced expression of IL-20 or dysregulated IL-23 signalling [107,110]. It remains an open question whether these findings are clinically important. In line with this, it is not clear if blocking this part of cytokine signalling may benefit patients with systemic sclerosis. 

The therapeutic potential of JAKi may also be explained in another way. Systemic sclerosis is characterized by the overexpression of interferon type 1 [76,111,112]. This pathophysiological phenomenon, commonly referred to as an interferon signature, is characterized by the overexpression of several proinflammatory interferon-related genes. The type 1 group of interferons comprises a group of five classes of interferons, namely, α, β, ω, ε and κ. All types of INT T1 classes signal through the same type 1 IFN heterodimeric receptor complex coupled with the JAK kinases. This brings novel therapeutic possibilities, as inhibiting JAK can block the whole interferons’ signalling. Blocking the interferon signalling seems to be relatively safe. According to a phase I study, the administration of MEDI-546, a human immunoglobulin G1 kappa monoclonal antibody directed against IFNAR1, resulted in the sustained inhibition of the type 1 IFN gene signature [113]. Following this promising result, a study with anifrolumab (anti-IFNAR1 monoclonal antibody) showed the reduced suppression of the IFN signature and TGFβ signalling in SSc skin [114]. This shows that blocking the IFN receptor might be a promising way to reduce disease activity in subjects with SSc or other IFN-related inflammatory diseases. Whether this finding translates to the inhibition of IFN signalling via JAK inhibitors should be elucidated. 

Recent analysis of case reports and small uncontrolled trials has shown the clinical benefit of JAK inhibitors in interstitial lung diseases associated with SSc. Despite many limitations from these studies, the results indicate that targeting several cytokines with one drug may be a promising way to halt the progression of fibrosis and inflammation [115]. 

## 7. Autoantibody-Targeted Therapy

The prevalence of antinuclear antibodies (ANA) in SSc is almost universal, with ANA detected in the sera of more than 90% patients with the disease [44]. Among the specific antibodies identified are anti-centromeric protein antibodies (CENP) topoisomerase I (SCl-70) and anti RNA polymerase III, which are commonly used for classification and prognosis staging and are linked with specific organ involvement [116]. 

Other targets for autoantibodies in SSc are endothelial cells, vascular smooth cells and fibroblasts. The interaction between specific autoantibodies and the targeted group of cells results in the activation, damage, and expression of a pro-fibrotic and pro-adhesive phenotype [117,118]. Indeed, specific autoantibodies have been identified, and their activity was likened to a pathogenic process, such as myofibroblasts transformation and direct endothelial damage, that leads to vascular injury [35,119,120,121].

The presence of antinuclear and more general autoantibodies is a hallmark of the disease, linked pathogenetically with the disease subtype course and prognosis. Therefore, therapeutic strategies aimed at reducing the generation of autoantibodies (currently tested in other autoimmune diseases) seems to be a reasonable approach in the treatment of SSc. 

The first choice to reduce autoantibody formation is to target the source of antibodies, the B cells. This approach was addressed in detail in the chapter on the role of B cells in systemic sclerosis.

However, there are plenty of opportunities that are beyond the structure and function of B cells.

The fate and function of B cells is precisely regulated by B cell regulatory factors. Among many cytokines, chemokines, and stimulatory factors affecting the structure and function of B cells, BAFF, also called B lymphocyte stimulator (BLyS), and APRIL, which is a homologous factor to BAFF, are recognized as key elements to regulate B cell activity. BAFF is expressed by monocytes, macrophages, and activated T cells. BAFF can interact with three receptors, the BAFF-R, BCMA, or TACI, and regulates B cell survival, differentiation, maturation, immunoglobulin class switching, and antibody production [122]. Following this understanding of the function of BAFF/APRIL, a phase II study investigating the effects of a combination of belimumab and mycophenolic acid in 20 patients with dcSSc was performed. Unfortunately, patients who received Belimumab did not benefit from an improvement in skin thickness compared to the placebo group [123]. 

B cell-targeting therapy is currently licensed in the treatment of rheumatoid arthritis and ANCA vasculitis but is also successfully used off-label in other immune-mediated diseases [124,125,126]. Unfortunately, most B cell-depleting therapies target the CD20 molecule expressed on the surface of the majority of B cells. With the progress of B cell maturation, the expression of this molecule reduces, and finally CD20 is absent on the surface of the most-differentiated B cell populations, making impossible to reduce the fully differentiated B cell population. To overcome this, plasma cells, a main site of autoantibody synthesis, can be affected from “inside”. This strategy is currently being realized with the use of proteasome inhibitors, a class of drug which is able to stop the proteasome activity that results in the accumulation of defective immunoglobulin chains, misfolded proteins, and the overflow of the endocytoplasmatic reticulum and directly leads to plasma-cell apoptosis [127,128,129].

In in vitro studies, proteasome inhibitors showed the potential to reduce collagen synthesis by human fibroblasts. So far, this mechanism was not confirmed in a mouse model of SSc, which showed no effect on skin and lung fibrosis [130].

Data from current phase II studies assessing a combination of Bortezomib, a protease inhibitor, and mycophenolate mofetil in patients with SSc, suggested that combination therapy was superior over Bortezomib alone (NCT02370693). A second-generation proteasome inhibitor Ixazonib is about to be tested in another trial (NCT04837131).

### 7.1. Bruton’s Kinase Inhibitors

Bruton’s tyrosine kinases (BTK) are members of the Tec kinases family of tyrosine protein kinases. BTK is a component of multiple signalling pathways and plays an important role in B-cell receptor (BCR) signalling, and in the development and activation of B cells [131,132].

The idea to use BTK inhibitors in systemic sclerosis is generally based on the observation of the role of B cells (and myeloid cells) in the development of SSc [36,43].

In a murine model of systemic sclerosis, BTKB66 (a new BTK inhibitor) treatment resulted in a reduction in skin fibrosis, collagen deposition, and decreased inflammatory cell infiltration in the skin [133]. In another ex vivo study testing the influence of Ibrutinib, a first-in-class, irreversible inhibitor of BTK, it was able to reduce the release of proinflammatory IL-6 and TNF-α. At the concentrations studied, Ibrutinib also reduced the concentration of profibrotic cytokines without any significant influence on IL-1 and IFN-γ [134].

Currently, 13 BTK-inhibitor drug candidates are being evaluated in clinical trials (phase II or higher) in autoimmune diseases. Unfortunately, the studies started so far focus mainly on systemic lupus erythematosus, rheumatoid arthritis, and multiple sclerosis [135]. 

### 7.2. Targeting CD38 Molecule

The other therapeutic approach which aims to reduce the concentration and formation of autoantibodies in SSc may involve the targeting of plasma cells and memory B cells. Moreover, recent data showed an elevated expression of CD38 in peripheral blood plasmablasts and plasma cells in SSc patients, suggesting a pathogenic role of this molecule in the development of the disease. This also suggests that CD38 may be a potential target to reverse the immunological abnormalities in the disease [136]. Taking into account that short-lived plasmablasts and plasma cells, and long-lived plasma cells, are important sources of the generation of autoantibodies, targeting the population of antibody-producing cells might be a reasonable approach [137,138,139].

Targeting the CD38 molecule may also bring other therapeutic effects in systemic sclerosis. The CD38 molecule represents the NADase enzyme, a type II plasma-membrane protein expressed on both immune and non-immune cells [140]. CD38 shows hydrolytic activity against nicotinamide adenine dinucleotide (NAD^+^), and its main role is to break down the NAD and reduce its level in the tissues. In the course of SSc, CD38 expression is elevated and correlates with disease activity and skin involvement. Thus, the negative modulation of CD38 function may translate into the restoration of NAD^+^ homeostasis in fibroblasts, affect myofibroblast activation, halt profibrotic cellular signalling, suppress the fibrotic gene, and restore endothelial function [141,142]. 

### 7.3. Mechanical Removal of Autoantibodies

This term refers to many procedures in which circulating autoantibodies are mechanically removed from the blood. The most commonly used procedures are therapeutic plasma exchange, plasmapheresis, rheoperesis, and immunoadsorption. Data on the therapeutic potential of these procedures come from small studies and case reports, generally reporting favourable therapeutic effects on skin fibrosis, musculoskeletal symptoms, Raynaud’s phenomenon, the healing of digital ulcers and organ manifestation. These techniques are reviewed in detail elsewhere [143].

Finally, the other approach that has been proposed recently is the epigenetic modulation of DNA methylation, histone modifications and non-coding RNAs in SSc [144]. Obviously, the role of this sophisticated approach in a real-world SSc clinic should be elucidated.

## 8. Conclusions

For many years, SSc was a disease without any treatment. However, developments in rheumatology and clinical immunology are resulting in the understanding of the key mechanisms responsible for driving inflammation and fibrosis in SSc. In line with this, several targets for novel drugs have been proposed (Figure 2).

Abatacept, Acazicolcept and Brentuximab act in the early phase of immunocompetent cell cross-talks, resulting in immune tolerance or immunosuppression. Targeting B cell function directly with anti-CD20 (Rituximab), antiCD22 (Epratuzumab), or anti-CD38 (Daratumumab) antibodies promotes B cell apoptosis, halting their transformations to mature plasmablasts, and, thus, preventing autoantibody synthesis. With the use of Janus kinase inhibitors, a reduction in the whole spectrum of class I and class II cytokines is observed, which potentially translates to halting the inflammatory response (early phase of the disease) or fibrosis (late stage of the disease). The same may be achieved with the use of specific antagonists of selected cytokines or its receptors (potent profibrotic IL-4 and IL-13 or interferons). Additionally, targeting plasma cells with proteasome inhibitors results in a reduction in autoantibody synthesis. Finally, the reduction in autoantibody levels can be achieved by mechanically removing circulating antibodies with the use of total plasma exchange, rheopheresis, plasmapheresis or antibody absorption.

With the heterogeneity seen in the pathophysiological background and clinical presentation, it seems unlikely that we will discover one drug that works sufficiently in all patients with SSc. However, a detailed assessment of the clinical course may support interventions appropriate for each individual. This seems to be a reasonable approach to distinguish between the early (inflammation) and late phases (fibrosis) of the disease and to try to match the disease presentation with the required therapeutic procedure.

## Figures and Tables

**Figure 1 pharmaceuticals-16-01066-f001:**
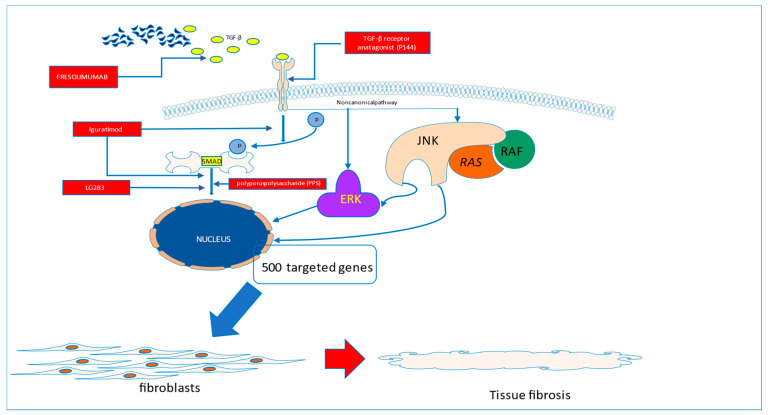
Fibrotic pathways to be blocked in systemic sclerosis.

**Figure 2 pharmaceuticals-16-01066-f002:**
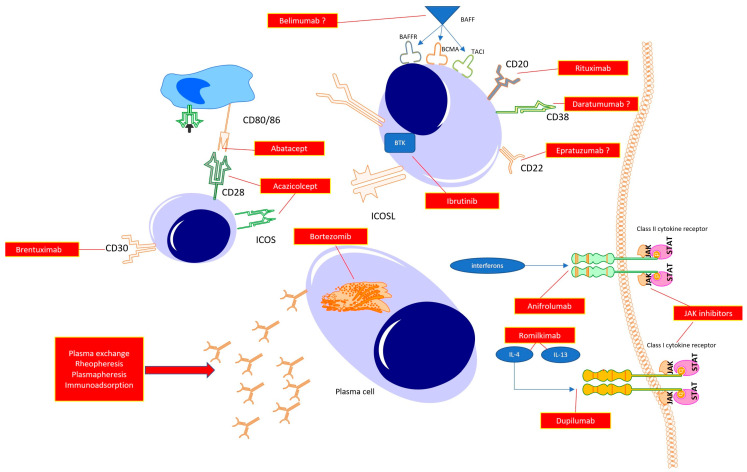
Possible therapeutic targets in the treatment of systemic sclerosis.

## Data Availability

Publicly available datasets were analysed in this study. This data can be found at the Pubmed database (https://www.ncbi.nlm.nih.gov/, accessed on 1 January 2023).

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
