# Peer review of "Novel Therapeutic Strategies in the Treatment of Systemic Sclerosis"

_pharmaceuticals, 2023, doi:10.3390/ph16081066_

Round 1

Reviewer 1 Report

In this review, the authors discuss potential therapeutic options designed to halt the progression of fibrotic processes, restore abnormal immune response, halt improper signaling from proinflammatory cytokines, and stop the production of disease-related autoantibodies. Manuscript can be accepted after following revisions.

a- Provide the reference for following statements:

First attempts to systematically assess the utility of dozens of various compounds for SSc treatment started in the 1960s. Since that time it was apparent that currently no specific drug able to halt disease progression exists. Therefore special emphasis was put on treating scleroderma-related internal organ involvement and this philosophy is currently used for the treatment.

EULAR recommendations for treatment of systemic sclerosis published in 2017 address the therapeutic approaches in several SSc-related organ complications such as vascular disease (Raynaud's phenomenon, digital ulcers ) pulmonary arterial hypertension (PAH), skin fibrosis, interstitial lung disease, scleroderma renal crisis and gastrointestinal involvement.

At the current level of understanding, the fibrotic process is the direct result of aberrant immunocompetent cell function, however, this process is directly mediated by several “fibrotic messengers“ and cytokines activating a plethora of profibrotic pathways.

b- All sections should be properly numbered and divided in sub sections.

 make a number or bullet for "EULAR recommendation for treatment patients with SSc" on page 2.

c-Figure is not good idea in conclusion section, please insert at anyother suitable pleace.

There are various formatting and section division issues and some sentences are so long without fluency.

Author Response

Thank you very much for your comments. According to Your suggestions, we did some changes. Please find the list of changes we made.

a- Provide the reference for the following statements:

First attempts to systematically assess the utility of dozens of various compounds for SSc treatment started in the 1960s. Since that time it was apparent that currently no specific drug able to halt disease progression exists. Therefore special emphasis was put on treating scleroderma-related internal organ involvement and this philosophy is currently used for the treatment.

The following citations were added

  1. Black, C.M. Systemic sclerosis: Is there a treatment yet? Annals of the rheumatic diseases 1990, 49, 735-737.
  2. Fernández-Codina, A.; Walker, K.M.; Pope, J.E. Treatment algorithms for systemic sclerosis according to experts. Arthritis & rheumatology (Hoboken, N.J.) 2018, 70, 1820-1828.

EULAR recommendations for treatment of systemic sclerosis published in 2017 address the therapeutic approaches in several SSc-related organ complications such as vascular disease (Raynaud's phenomenon, digital ulcers ) pulmonary arterial hypertension (PAH), skin fibrosis, interstitial lung disease, scleroderma renal crisis and gastrointestinal involvement.

  1. Kowal-Bielecka, O.; Fransen, J.; Avouac, J.; Becker, M.; Kulak, A.; Allanore, Y.; Distler, O.; Clements, P.; Cutolo, M.; Czirjak, L., et al. Update of eular recommendations for the treatment of systemic sclerosis. Annals of the rheumatic diseases 2017, 76, 1327-1339.

At the current level of understanding, the fibrotic process is the direct result of aberrant immunocompetent cell function, however, this process is directly mediated by several “fibrotic messengers“ and cytokines activating a plethora of profibrotic pathways.

  1. Bhattacharyya, S.; Wei, J.; Varga, J. Understanding fibrosis in systemic sclerosis: Shifting paradigms, emerging opportunities. Nature reviews. Rheumatology 2011, 8, 42-54.
  2. Mouawad, J.E.; Feghali-Bostwick, C. The molecular mechanisms of systemic sclerosis-associated lung fibrosis. International journal of molecular sciences 2023, 24, 2963.

b- All sections should be properly numbered and divided in sub sections.                                                                                                                

We subdivide the section into subsections  with proper headlines for better understanding/reading

 make a number or bullet for "EULAR recommendation for treatment patients with SSc" on page 2.

All recommendations are preceded by bullets as requested

c-Figure is not good idea in conclusion section, please insert at anyother suitable pleace.

Figure moved to the central part of the manuscript

Comments on the Quality of English Language

There are various formatting and section division issues and some sentences are so long without fluency.

With the help of my friend English native speaker professor of Oxford University, we fixed grammar and spelling problems

Reviewer 2 Report

In this narrative review, the Authors aimed to summarize the novel therapeutic strategies being evaluated in systemic sclerosis.

Although the topic is of interest, the manuscript as it is does not provide a comprehensive and unbiased overview of what is important to know in recent advances in SSc treatment.

MAJOR

1. The pathogenesis of SSc includes fibrotic, immunological and vascular processes. The latter is not considered at all in the review, though vascular impairment constitutes an important event in SSc pathogenesis and a substantial clinical burden for the patients.

2. Apart from including vascular targets, I would also join together the paragraphs on cellular immunity and autoantibody. Indeed, autoantibodies are a hallmark of the disease but their pathogenicity is yet to be established. Thus, it is difficult to distinguish if the targeting of B cells and plasma cells works for reduced autoantibody production or because of reduced APC activity or cytokine production etc.

3. Several important anti-fibrotic therapies being evaluated in SSc have been neglected, such as the Oncostatin M inhibitors, Rho-associated kinase inhibitors, melanocortin agonists, AVID2000, etc. 

4. Cellular therapies (i.e. mesenchymal stromal cells, CAR-T, etc) aimed to restore the activity of immune system should be cited as there are some studies supporting their use.

5. Finally, the manuscript is too long and difficult to read. It would benefit from improved internal organization, with sub-paragraphs etc.

MINOR

1. The term "generalized" SSc is not commonly used. Use "limited cutaneous" and "diffuse cutaneous" instead.

2. Phase III studies with Rituximab not cited by the Authors have been published recently, such as the DESIRES trial and the RECITAL trial.

Extensive spell and punctuation checks are needed.

Author Response

We appreciate your comments please find the explanation and proposals for changes

MAJOR

  1. The pathogenesis of SSc includes fibrotic, immunological and vascular processes. The latter is not considered at all in the review, though vascular impairment constitutes an important event in SSc pathogenesis and a substantial clinical burden for the patients.

We entirely agree that more data should be given therefore we added the special subsection  dedicated entirely to vasculopathy has been added

Vasculopathy

This term was almost exclusively used for the description of small anrge vessel abnormalities in SSc[30]. This process is commonly recognized as the third (among fibrosis and immune system dysfunction)  key element in the pathogenesis of systemic sclerosis. Vascular injury is believed to be initiated by the influence of environmental factors and autoimmune responses directed toward endothelial cells[31]. The latter is caused mainly by anti-endothelial antibodies.  Vascular abnormalities seen in SSc are classified into two categories 1. Destructive vasculopathy and 2. impaired vasculogenesis and angiogenesis. In the early stages of SSc predominant activity of inflammatory cells is observed that leads directly to activation of endothelial cells[32]. Parallel to this increased synthesis of pro-angiogenic factors (mainly VEGF and endothelin-1) is observed despite the impaired response of ECs to those factors. At the functional level, vasculopathy is characterized by excessive reactive oxygen species formation, infiltration of Th cells polarized, toward  Th17 response, local infiltration of macrophages, activation of endothelial to mesenchymal transition promoting vascular remodeling and tissue fibrosis[33].

  1. Apart from including vascular targets, I would also join together the paragraphs on cellular immunity and autoantibody. Indeed, autoantibodies are a hallmark of the disease but their pathogenicity is yet to be established. Thus, it is difficult to distinguish if the targeting of B cells and plasma cells works for reduced autoantibody production or because of reduced APC activity or cytokine production etc.

We are entirely agree  therefore we  made some explanation to it  and stated

The role of B cells is not, however, restricted to the production of scleroderma-related antibodies, which seem to play a rather indicative but not causative role in SSc[35]

  1. Several important anti-fibrotic therapies being evaluated in SSc have been neglected, such as the Oncostatin M inhibitors, Rho-associated kinase inhibitors, melanocortin agonists, AVID2000, etc.

We added more data on the theratemet strategies pointed by the Reviewer 

The other anti-fibrotic treatments

Oncostatin M

Oncostatin M (OSM) is a member of the IL family and signaling through JAK/STA system. OSM levels were found to be elevated in the serum of SSc patients, moreover skin of SSc patients is characterized by the high expression of OSM receptor suggesting the pathophysiological role of this cytokine in the development of the disease[25,26]. Unfortunately recently finished trial with GSK233081- a humanized IgG1 antibody targeting the OSM receptor failed to show any therapeutic effect in SSc[27].

Rho-associated kinases

The other potential anti-fibrotic therapeutic approach is to modulate the Rho-associated kinase activity. Rho kinases (ROCK) are essential downstream effectors of the  Rho GTPases and are responsible for important physiological functions such as the organization of actin cytoskeletal cell cycle control, apoptosis, and regulation of cell-cell adhesion[28]. It was established that ROCK are crucial  for myofibroblast differentiation and synthesis of extracellular matrix thus might be a potential therapeutical target for SSc-related fibrosis. In line with it Belumosudil a ROCK inhibitor licenses for the treatment chronic graft-versus-host diseases is under investigation for systemic sclerosis in the USA (NCT02841995 and NCT04930562)[29].

  1. Cellular therapies (i.e. mesenchymal stromal cells, CAR-T, etc) aimed to restore the activity of immune system should be cited as there are some studies supporting their use.

In the chapter immunocompetent cells in systemic sclerosis we providea some data on CAR-T stated that

Recently a case report on CD19-CAR T cells showed promising data on utility of such a treatment in patient with severe systemic sclerosis[38].

  1. Finally, the manuscript is too long and difficult to read. It would benefit from improved internal organization, with sub-paragraphs etc.

We did several changes in the structure and organisation of the manuscript mire subheading were added all changes can be seen as the are underlined by changes tracking system

MINOR

  1. The term "generalized" SSc is not commonly used. Use "limited cutaneous" and "diffuse cutaneous" instead.

Generalized changed to diffuse as proposed

  1. Phase III studies with Rituximab not cited by the Authors have been published recently, such as the DESIRES trial and the RECITAL trial.

We decided to add data on mentioned studies as proposed

Recently the results from phase with rituximab  (RECITAL and DESIRES) showed promising results for skin and lung fibrosis improvement[42,43], , Discussing available data however a key question remains open, whether results from these studies will be confirmed in larger phase III trials?

Comments on the Quality of English Language

Extensive spell and punctuation checks are needed

We did extensive review of the manuscript in terms of grammar and punctuation We appreciate the help of our friend professor of Oxford University

Reviewer 3 Report

It is clearly written and understandable for a lay person in basic science

minor typo:

- line 112 TBF beta rather than TGH

- inconsistency in abbreviations: sometimes 'beta' sometimes greek letter

Author Response

Thank you very much for you review, we appreciate your commnets

  • line 112 TBF beta rather than TGH

- inconsistency in abbreviations: sometimes 'beta' sometimes greek letter

All necessary corrections done

Round 2

Reviewer 1 Report

Authors have improved the manuscript as advised.

Reviewer 2 Report

The Authors addressed the suggestions. I have no further comments.

Some minor revisions are needed